# Learning dynamics of deep linear networks with multiple pathways

**Jianghong Shi**
Department of Applied Mathematics
University of Washington
Seattle, WA 98195
jhshi@uw.edu

**Eric Shea-Brown**
Department of Applied Mathematics
University of Washington
Seattle, WA 98195
etsb@uw.edu

**Michael A. Buice**
Allen Institute MindScope Program
Seattle, WA 98109
michaelbu@alleninstitute.org

## Abstract

Not only have deep networks become standard in machine learning, they are increasingly of interest in neuroscience as models of cortical computation that capture relationships between structural and functional properties. In addition they are a useful target of theoretical research into the properties of network computation. Deep networks typically have a serial or approximately serial organization across layers, and this is often mirrored in models that purport to represent computation in mammalian brains. There are, however, multiple examples of parallel pathways in mammalian brains. In some cases, such as the mouse, the entire visual system appears arranged in a largely parallel, rather than serial fashion. While these pathways may be formed by differing cost functions that drive different computations, here we present a new mathematical analysis of learning dynamics in networks that have parallel computational pathways driven by the same cost function. We use the approximation of deep linear networks with large hidden layer sizes to show that, as the depth of the parallel pathways increases, different features of the training set (defined by the singular values of the input-output correlation) will typically concentrate in one of the pathways. This result is derived analytically and demonstrated with numerical simulation with both linear and non-linear networks. Thus, rather than sharing stimulus and task features across multiple pathways, parallel network architectures learn to produce sharply diversified representations with specialized and specific pathways, a mechanism which may hold important consequences for codes in both biological and artificial systems.

## 1 Introduction

Deep networks are increasingly used as models of cortical computation, particularly of the visual pathway. Indeed, the earliest convolutional neural network was a model of the mammalian visual system [6] and the use of CNNs to model the visual pathway has continued [14, 21, 22, 3, 2]. These models are usually serial models of the primate ventral stream that describe the computation of that system as being closely related to object recognition. However, processing of visual information in the primate is associated with (at least) two parallel pathways, the "ventral" (or "what") and the "dorsal" (or "where") pathways [4, 7, 10]. In the mouse visual system, anatomical analysis also suggests that there are parallel pathways [8, 19, 20], although the specific functional role of these

36th Conference on Neural Information Processing Systems (NeurIPS 2022).

pathways is less clear. Interestingly, recent research [1] has shown that self-supervised training in neural network architectures with parallel pathways can lead to the emergence of ventral-like and dorsal-like pathways. Another study shows that, with a similar self-supervised training objective, shallower architectures with parallel pathways lead to closer matches with functional data in the Allen Brain Observatory than deep single pathway architectures [11]. Furthermore, an anatomically constrained deep neural network model for the mouse visual cortex has demonstrated that such parallel architectures produce more diverse representations of visual images, compared to single pathway architectures [18].

Additionally, a coincident observation has been made for the original AlexNet [9] (a deep neural network designed for image processing) that also shows the emergence of functional specialization across parallel pathways, with different pathways (defined in the model because it was implemented on multiple GPUs) learning different features of the input data. Overall, in the road map for developing future artificial intelligence systems, architectures with parallel pathways promise potential solutions for multi-tasking, multi-sensory, energy-efficient computing [15]. But despite the richness of information representation and the potential capabilities of the architectures with parallel pathways, there is a lack of research in understanding the fundamental learning behavior that occurs in such architectures. Here, we advance this frontier through a mathematical analysis of the behaviour of deep neural networks with parallel pathways, in the tractable setting of deep linear networks [16].

Our starting point follows Saxe et al. [17], which elegantly shows how, in such a linear network, the training data and learning dynamics for a given task can be fully described via a set of independent modes resulting from singular value decomposition (SVD) of the input-output correlation matrix. During gradient decent, a deep linear network will pick up the task related information from the data by acquiring knowledge from each mode [17]. Here, we use the same overall framework to study the dynamics of learning in linear networks with parallel pathways.

Our analysis and simulations yield the following main results. For linear networks with multiple parallel pathways of hidden neurons that feed into the same output representation and cost function:

1. For wide networks with random weight initialization, the learning dynamics reduces to a coupled set of ordinary differential equations of dimension equal to the number of pathways for each non-zero singular vector in the input-output correlation matrix of the training set.

2. These dynamics are competitive in the sense that each pathway will compete to "explain" the singular values of the training set.

3. For deeper networks, this competition becomes more severe, resulting in more trajectories in phase space flowing to extreme solutions, wherein features (defined by the singular values of the training set) are confined largely to a single pathway.

4. For the strict infinite-width limit, the system flows to a state wherein features are shared evenly across the pathways. At finite width, fluctuations in the initial state put the system onto a trajectory leading to a biased solution, where the bias is more severe at larger depths. Thus in large-width, large-depth networks, the system evinces a finite-size induced spontaneous symmetry breaking.

We demonstrate this result in simulations and show empiricially that the main result holds in networks with non-linearities. Thus, this depth-dependent mechanism reveals a dynamical link between network architecture and representation learning.

## 2 Mathematical framework

**Network description:** We study the learning dynamics under gradient descent of a system with parallel pathways with one or more hidden layers in each pathway. We consider "linear" networks in which the transformation of each layer to the next is given by a matrix multiplication, with no non-linearity (i.e., linear transfer functions at each "node"). Given an input vector $x$ (of dimension $l_x$) and output vector $y$ (of dimension $l_y$), the output of the system with $M$ parallel pathways is given by

$$
\begin{aligned}
y &= \left( \sum_{a=1}^{M} \mathbf{W}_a^{D_a} \cdots \mathbf{W}_a^2 \mathbf{W}_a^1 \right) x \\
&= \mathbf{\Omega} x
\end{aligned}
\tag{1}
$$

where we have defined

$$\mathbf{\Omega} = \sum_{a=1}^{M} \mathbf{\Omega}_a \quad \equiv \quad \sum_{a=1}^{M} \prod_{d=1}^{D_a} \mathbf{W}_a^d \tag{2}$$

and where pathway $a$ has $D_a - 1$ hidden layers and $D_a$ weight matrices. Each weight matrix $\mathbf{W}_a^d$ has shape $N_a^d \times N_a^{d-1}$ (where $N_a^0 = l_x$). In the product formula, we are assuming that matrices with a higher value of $d$ are to the *left* of matrices with smaller values of $d$. A schematic of this system is given in Figure 1.

As described below, we consider training this network using input-output vector pairs $x$ and $y$ that have correlation matrix $\mathbf{\Sigma}^{yx} = \mathbf{USV}^T$, where we have used the singular value decomposition where $\mathbf{S}$ is diagonal and $\mathbf{U}$, $\mathbf{V}$ are orthogonal matrices. The gradient descent dynamics will lead to solutions $\mathbf{U}^T \mathbf{\Omega V} = \mathbf{S}$, which implies that in the steady state the pathway matrices $\mathbf{K}_a \equiv \mathbf{U}^T \mathbf{\Omega}_a \mathbf{V}$ must sum to the singular values. A schematic of this decomposition is shown in Figure 2(top row), adapted from [17].

**Main question and illustration of main result:** Given this description of how the network as a whole carries information about each input-output mode, we ask how this mode information is distributed across parallel pathways. We illustrate two possibilities in Figure 2: (1) each mode's information could be shared across multiple pathways (middle row) or (2) each mode's information could be exclusively carried in individual pathways (bottom row). We show that case (2) occurs in the limit of large network size and large depth due to finite-size induced spontaneous symmetry breaking.

## 2.1 Learning dynamics under gradient descent

Using the same setting as in Saxe et al. [16], we train the network with a set of $P$ examples $\{x^i, y^i\}, i = 1, 2, \ldots, P$ with gradient descent on the squared loss

$$L = \frac{1}{2} \sum_{i=1}^{P} ||y^i - \mathbf{\Omega}x^i||^2.$$

In the limit of small learning rate, the dynamics of gradient descent are approximated by the continuous time ordinary differential equations given below (for each pathway $a$):

$$\tau \frac{d}{dt} \mathbf{W}_a^d \quad = \quad \left( \prod_{i=d+1}^{D_a} \mathbf{W}_a^i \right)^T [\mathbf{\Sigma}^{yx} - \mathbf{\Omega\Sigma}^x] \left( \prod_{i=1}^{d-1} \mathbf{W}_a^i \right)^T \tag{3}$$

where $\prod_{i=d_1}^{d_2} \mathbf{W}_a^i = \mathbf{W}_a^{l_2} \mathbf{W}_a^{l_2-1} \cdots \mathbf{W}_a^{l_1}$ when $l_1 \leq l_2$ and $\prod_{i=l_1}^{l_2} \mathbf{W}_a^i = I$ when $l_1 > l_2$. $\mathbf{\Sigma}^{yx} = E[yx^T]$ and $\mathbf{\Sigma}^x = E[xx^T]$. We assume without loss of generality that $\mathbf{\Sigma}^x = \mathbf{I}$.

Define the initial conditions for the weight matrices as $\mathbf{R}_a^d \equiv \mathbf{W}_a^d(0)$. We take the components $\mathbf{R}_a^d$ to be independently and identically distributed according to a normal distribution with zero mean and variance $\frac{\sigma^2}{N_a^d}$, where $N_a^d$ is the number of units in the output of that layer. We also denote the singular value decomposition of $\mathbf{\Sigma}^{yx} = \mathbf{USV}^T$, where $\mathbf{S}$ is diagonal and $\mathbf{U}$, $\mathbf{V}$ are orthogonal matrices.

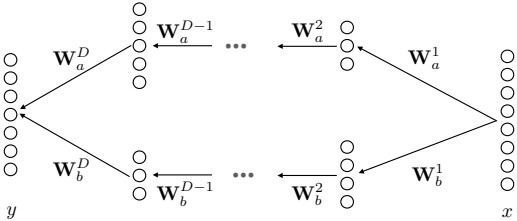

Figure 1: **Schematic of a multi-pathway network.** The first layer input feeds into multiple hidden layers in parallel, each of which a pathway through multiple hidden layers. The outputs of the final layer in each pathway linearly combine into the final output of the network.

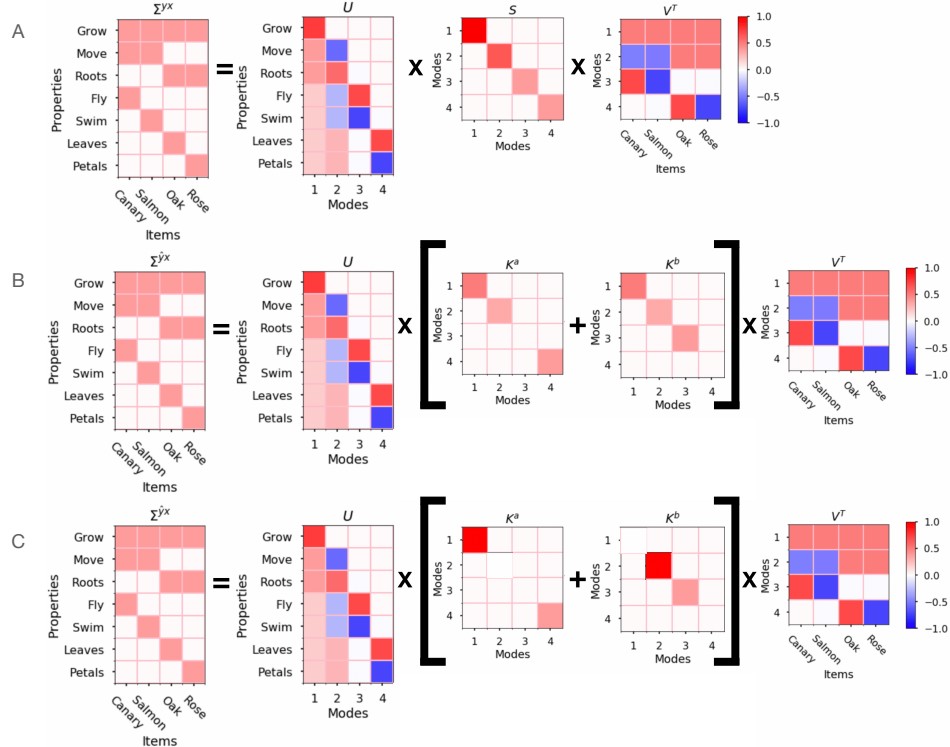

Figure 2: **Example of multi-pathway learning.** A) Example reconstructed from [17] showing the singular value decomposition of a hypothetical training set relating objects to properties. In the steady state, the network will learn the feature associations defined by the singular values. B) An example of learning in a shallow, two pathway network. In this case, the features will be arbitrarily split across the two pathways. C) An example of multi-pathway learning in a deep network. In this case, the feature learning will typically concentrate into one of the pathways for each feature.

## 3 Gradient descent solutions for large hidden layer size

Here we construct solutions to the equations (3). The general strategy is to choose coordinate systems for the neural representations in the vector space defined by the singular value decomposition. We then show that the solutions to the gradient descent equations (3) are diagonal in these coordinates.

### 3.1 Choosing orthogonal coordinates

Each hidden layer $d$ in each pathway $a$ provides an $N_a^d$ dimensional representation of the input space from which the vectors $x$ are derived. In order to construct solutions of the gradient descent equations of motion (3), we will choose coordinates on these representations that map to the space of singular values of the input-output correlation matrix $\mathbf{\Sigma}^{yx}$. We construct these coordinates using the initial value matrices $\mathbf{R}_a^d$. Note that for large $N_a^d$, we must have

$$\left(\mathbf{R}_a^d\right)^T \mathbf{R}_a^d \quad \approx \quad \sigma^2 \mathbf{I} \tag{4}$$

which is exact as $N_a^d \to \infty$. We use the matrices $\mathbf{U}$ and $\mathbf{V}$ to project the initial and final weight matrices onto the singular vectors: $\mathbf{W}_a^1 \to \mathbf{W}_a^1 \mathbf{V}$ and $\mathbf{W}_a^{D_a} \to \mathbf{U}^T \mathbf{W}_a^{D_a}$. In addition, we use the initial state matrices $\mathbf{R}_a^d$ to define a coordinate system in the singular value space for the intermediate representations. We define this map as follows:

$$\bar{\mathbf{W}}_a^1 \quad = \quad \sigma^{-2} \mathbf{V}^T \left(\mathbf{R}_a^1\right)^T \mathbf{W}_a^1 \mathbf{V} \qquad \text{if } d = 1 \tag{5}$$

$$\bar{\mathbf{W}}_a^d \quad = \quad \sigma^{-2d} \mathbf{V}^T \left(\mathbf{R}_a^1\right)^T \cdots \left(\mathbf{R}_a^d\right)^T \mathbf{W}_a^d \mathbf{R}_a^{d-1} \cdots \mathbf{R}_a^1 \mathbf{V} \qquad \text{if } 1 < d < D_a \tag{6}$$

$$\bar{\mathbf{W}}_a^{D_a} \quad = \quad \sigma^{-2D_a} \mathbf{U}^T \mathbf{W}_a^{D_a} \mathbf{R}_a^{D_a-1} \cdots \mathbf{R}_a^1 \mathbf{V} \qquad \text{if } d = D_a \tag{7}$$

These coorinate transformations imply the following relations:

$$\mathbf{V}^T \left(\mathbf{R}_a^1\right)^T \cdots \left(\mathbf{R}_a^{d-1}\right)^T \prod_{i=1}^{d-1} \mathbf{W}_a^i \mathbf{V} \;=\; \prod_{i=1}^{d-1} \bar{\mathbf{W}}_a^i \tag{8}$$

$$\mathbf{U}^T \left(\prod_{i=d+1}^{D_a} \mathbf{W}_a^i\right) \mathbf{R}_a^d \cdots \mathbf{R}_a^1 \mathbf{V} \;=\; \prod_{i=d+1}^{D_a} \bar{\mathbf{W}}_a^i \tag{9}$$

which together imply

$$\bar{\mathbf{\Omega}} = \mathbf{U}^T \mathbf{\Omega} \mathbf{V} = \sum_a \mathbf{K}_a = \sum_a \prod_{i=1}^{D_a} \bar{\mathbf{W}}_a^i \tag{10}$$

In the singular value coordinate space, the gradient descent equations of motion for each $a$ are now

$$\tau \frac{d}{dt} \bar{\mathbf{W}}_a^d \;=\; \left(\prod_{i=d+1}^{D_a} \bar{\mathbf{W}}_a^i\right)^T [\mathbf{S} - \bar{\mathbf{\Omega}}] \left(\prod_{i=1}^{d-1} \bar{\mathbf{W}}_a^i\right)^T \tag{11}$$

Note that in the steady state we have

$$\bar{\mathbf{\Omega}}(t \to \infty) = \mathbf{S} \tag{12}$$

In the initial state we have in the large $N_a^d$ limit:

$$\bar{\mathbf{W}}(0)_a^d = \mathbf{I} + \delta_a^d \qquad \text{if} d < D_a \tag{13}$$
$$\bar{\mathbf{W}}(0)_a^{D_a} = \delta_a^{D_a} \qquad \text{if} d = D_a \tag{14}$$

where $\delta_a^d \to 0$ in the large $N_a^d$ limit. So each $\bar{\mathbf{W}}_a^d$ is diagonal (or zero) in the initial state. By construction, these matrices are all of dimension $l_s \times l_s$, where $l_s$ is the number of singular values of $\mathbf{\Sigma}^{yx}$.

## 3.2  Diagonal solutions for the gradient descent equations

We propose the following ansatz for the solution:

$$\bar{\mathbf{W}}(t)_a^d = \mathbf{\Gamma}_a^d(t) + \bar{\delta}_a^d \tag{15}$$

where $\mathbf{\Gamma}_a^d(t)$ are all diagonal matrices, with diagonal entries given by the vector $\Gamma_a^d(t)$, with components $\left(\Gamma_a^d(t)\right)_\alpha$, where $\alpha$ is an index over the singular values. The terms $\bar{\delta}_a^d \to 0$ in the large $N_a^d$ limit. With this ansatz, in the $N_a^d \to \infty$ limit, we have for each pathway $a$ and component $\alpha$:

$$\tau \frac{d}{dt} \left(\Gamma_a^d\right)_\alpha \;=\; \prod_{i=d+1}^{D_a} \left(\Gamma_a^i\right)_\alpha [S_\alpha - \bar{\Omega}_\alpha] \prod_{i=1}^{d-1} \left(\Gamma_a^i\right)_\alpha \tag{16}$$

where $S_\alpha$ and $\bar{\Omega}_\alpha$ are the diagonal components of $\mathbf{S}$ and $\bar{\mathbf{\Omega}}$, respectively.

For each $\alpha$, each of the $\left(\Gamma_a^d\right)_\alpha$ for $d < D_a$ have the same initial conditions, so by symmetry we can define $q_{a\alpha} = \left(\Gamma_a^d\right)_\alpha$ for all $d < D_a$ and $p_{a\alpha} = \left(\Gamma_a^{D_a}\right)_\alpha$ to reduce these equations to

$$\tau \frac{d}{dt} q_{a\alpha} \;=\; q_{a\alpha}^{D_a - 2} p_{a\alpha} \left[S_\alpha - \bar{\Omega}_\alpha\right] \tag{17}$$
$$\tau \frac{d}{dt} p_{a\alpha} \;=\; q_{a\alpha}^{D_a - 1} \left[S_\alpha - \bar{\Omega}_\alpha\right]$$

for each pathway $a$ and component $\alpha$. The initial states of these variables are $q_{a\alpha} = 1$ and $p_{a\alpha} = 0$ in the large $N_a^d$ limit. Moreover we have $\bar{\Omega}_\alpha = \sum_{a=1}^M K_{a\alpha} = \sum_{a=1}^M p_{a\alpha} q_{a\alpha}^{D_a - 1}$. Importantly, these diagonal solutions are stable to off-diagonal perturbations, meaning that at finite size the small corrections $\bar{\delta}_a^d$ will decay to zero (see the section Simulations and Figure 4).

Taking the ratio of each of the equations in (17) above yields the differential equation

$$\frac{dq_{a\alpha}}{dp_{a\alpha}} = \frac{p_{a\alpha}}{q_{a\alpha}} \tag{18}$$

and thus these values are related by

$$p_{a\alpha}^2 = q_{a\alpha}^2 - 1 \tag{19}$$

The variables $p_{a\alpha}$ are thus sufficient to specify the solution and they obey the following equation:

$$\tau \frac{d}{dt} p_{a\alpha} = \left(\sqrt{p_{a\alpha}^2 + 1}\right)_\alpha^{D_a - 1} \left[S_\alpha - \bar{\Omega}_\alpha\right] \tag{20}$$

with $\bar{\Omega}_\alpha = \sum_{a=1}^M p_{a\alpha} \left(\sqrt{p_{a\alpha}^2 + 1}\right)^{D_a - 1}$. Different pathways will then be related by the relationship:

$$\frac{dp_{a\alpha}}{dp_{b\alpha}} = \frac{\sqrt{p_{a\alpha}^2 + 1}^{D_a - 1}}{\sqrt{p_{b\alpha}^2 + 1}^{D_b - 1}} \tag{21}$$

which gives

$$
\begin{aligned}
p_{a\alpha\,2}F_1\left(\frac{1}{2}, \frac{D_a - 1}{2}, \frac{3}{2}, -p_{a\alpha}^2\right) - p_{b\alpha\,2}F_1\left(\frac{1}{2}, \frac{D_b - 1}{2}, \frac{3}{2}, -p_{b\alpha}^2\right) &= \\
\bar{\delta}_{a\alpha\,2}F_1\left(\frac{1}{2}, \frac{D_a - 1}{2}, \frac{3}{2}, -\bar{\delta}_{a\alpha}^2\right) - \bar{\delta}_{b\alpha\,2}F_1\left(\frac{1}{2}, \frac{D_b - 1}{2}, \frac{3}{2}, -\bar{\delta}_{b\alpha}^2\right) &= C
\end{aligned}
\tag{22}
$$

where $_2F_1$ is the hypergeometric function and we have defined the constant $C$. We have re-introduced the finite size initial conditions via $\bar{\delta}_{a\alpha}$. These are determined by the random matrix initializations and will vanish in the large network size limit. In the strict limit, the solutions therefore must be equal across all pathways. Finite size fluctuations in the initial state will break this symmetry.

The phase planes for a two pathway network with common depths $D = D_1 = D_2$ is plotted for various values of $D$ in Figure 3. Each row represents the same set of 100 initial conditions drawn from zero mean normal distributions of a given variance, with the larger variance in the top row. Note that as the depth $D$ increases, so does the curvature of the trajectories away from zero. The sharper trajectories push the steady state solution to one where one or the other pathway dominates $\bar{\Omega}_\alpha = S_\alpha$, so one can see the result already in the phase portrait that at larger depths the contribution to $\bar{\Omega}_\alpha$ will come from a single pathway.

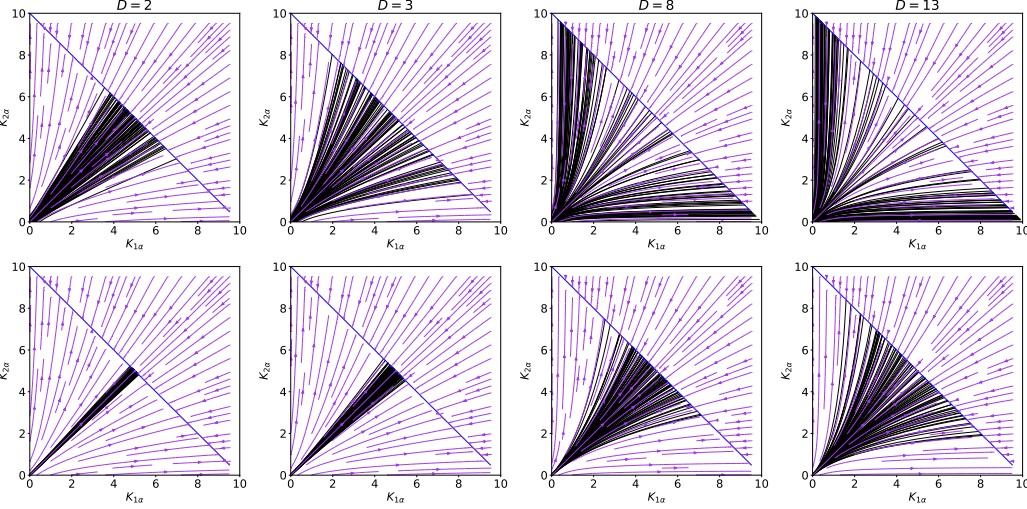

Figure 3: **Phase portraits for two pathway networks at different depths for a single** $\alpha$**.** $S_\alpha = 10, D_a = D_b$. Initial conditions are drawn from zero mean 2d Gaussians with $\sigma = 0.1$ (top) and $\sigma = 0.01$ (bottom). All initial values are the same for each row. The steady state is governed by $S_\alpha = K_{1\alpha} + K_{2\alpha}$, shown by the blue diagonal line. Notice that for deeper networks and larger initial fluctuations the trajectories tend towards more extreme values where one or the other pathway dominates for a fixed initial condition.

**Asymptotic approximation** While the result is already qualitatively clear from the phase portrait, we can obtain an analytic statement. For a more easily interpretable approach than analyzing hypergeometric functions (and a way of seeing the asymptotic behavior of the above equation), consider the case where $S_\alpha$ is sufficiently greater than 1 so that we can approximate

$$p_{a\alpha} \approx q_{a\alpha} \tag{23}$$

(One can check this with a post-hoc self consistency condition. For the above relation to hold, $S_\alpha$ will need to be large.) Then we have

$$\frac{dp_{a\alpha}}{dp_{b\alpha}} = \frac{p_{a\alpha}^{D_a-1}}{p_{b\alpha}^{D_b-1}} \tag{24}$$

which implies

$$\frac{1}{2-D_a} p_{a\alpha}^{2-D_a} - \frac{1}{2-D_b} p_{b\alpha}^{2-D_b} = C_{ab\alpha} \tag{25}$$

where $C_{ab\alpha}$ is a constant that depends upon initial conditions and which further implies

$$\frac{1}{2-D_a} (K_{a\alpha})^{2/D_a-1} - \frac{1}{2-D_b} (K_{b\alpha})^{2/D_b-1} = C_{ab\alpha} \tag{26}$$

When $D_a, D_b$ are large we have

$$\frac{1}{K_{a\alpha}} - \frac{1}{K_{b\alpha}} = C'_{ab\alpha} \tag{27}$$

for some $C'_{ab\alpha}$. In steady state, we have $S_\alpha = \sum_a K_{a\alpha}$. Consider $S'_\alpha = S_\alpha - \sum_{c \neq a,b} K_{c\alpha} = K_{a\alpha} + K_{b\alpha}$. Then in the large depth limit we can write for $K_{b\alpha}$

$$\frac{1}{S'_\alpha - K_{b\alpha}} - \frac{1}{K_{b\alpha}} = C'_{ab\alpha} \tag{28}$$

the solutions of which are given by the quadratic equation

$$C'_{ab\alpha} K_{b\alpha}^2 + (2 - S'_\alpha C'_{ab\alpha}) K_{b\alpha} - S'_\alpha = 0 \tag{29}$$

which are

$$K_{b\alpha} = \frac{C'_{ab\alpha} S'_\alpha - 2 \pm \sqrt{(C'_{ab\alpha} S'_\alpha - 2)^2 + 4 S'_\alpha C'_{ab\alpha}}}{2 C'_{ab\alpha}} \tag{30}$$

In the limit $|C'_{ab\alpha}| \to \infty$ and in the domain $K_{b\alpha} \in [0, S'_\alpha]$, these will be

$$K_{b\alpha} = \frac{S'_\alpha}{2} + \mathrm{sign}(C'_{ab\alpha}) \frac{S'_\alpha}{2} \tag{31}$$

and thus one pathway or the other dominates the singular value $\alpha$ for each $\alpha$, depending upon the sign of $C_{ab\alpha}$. As mentioned above, in the strict limit the constant $C_{ab\alpha} \to 0$, leaving symmetric solutions across the pathways. In the finite size limit, fluctuations in the initial state will drive $C_{ab\alpha}$ to larger values, biasing the network to one pathway or the other. We note that in a real application, initial conditions are not necessarily drawn from the distributions that we assume, and we expect effectively larger initial fluctuations than our initial state would imply, e.g. by choosing the initial values of each weight matrix from $N(0, \sigma)$, rather than scaling with network size (see the examples in Simulations). Such larger fluctuations would only enhance the effect of the symmetry breaking.

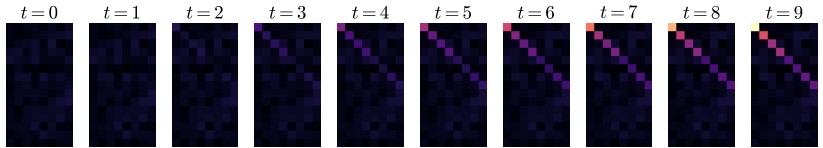

Figure 4: **Learning dynamics leads to diagonal representations in each pathway.** An example realization of learning in $\mathbf{K}_a$ for a single pathway. In the initial state the $\mathbf{K}_a$ starts as a uniform random matrix and quickly becomes diagonal after a few time steps.

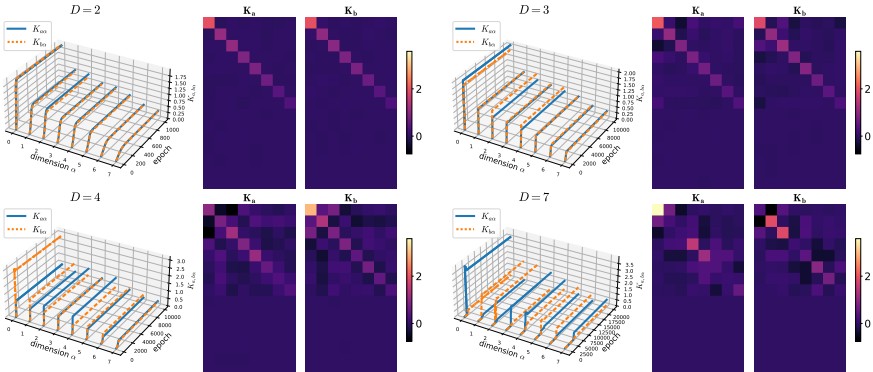

Figure 5: **Learning dynamics in two pathway networks with increasing depth.** $D_a = D_b = D$. Left to right, top to bottom are $D = 2, 3, 4$, and 7, respectively. The left side of each sub-panel shows trajectories of the matrices $\mathbf{K}_a$ and $\mathbf{K}_b$ in the singular value space. The right side of each sub-panel shows the final values for $\mathbf{K}_a$ and $\mathbf{K}_b$ in the singular value space. Note how with larger depths, the features have a stronger tendency to concentrate on one pathway or the other. $N_a = N_b = 1000$ and $\sigma = 0.01$.

# 4 Simulations

We demonstrate our results with numerical simulations of networks with two pathways and multiple depths. For these examples we use the same number of layers per pathway and $N_1 = N_2 = 1000$. The initial state of the weight matrices is drawn from a zero mean normal distribution with a fixed standard deviation $\sigma = 0.01$. The input vectors $x$ are 8-dimensional and are the rows of the 8-dimensional identity matrix. The output vectors $y$ are 15-dimensional and are the rows of the matrix:

$$Y = \begin{bmatrix} 1 & 1 & 0 & 1 & 0 & 0 & 0 & 1 & 0 & 0 & 0 & 0 & 0 & 0 & 0 \\ 1 & 1 & 0 & 1 & 0 & 0 & 0 & 0 & 1 & 0 & 0 & 0 & 0 & 0 & 0 \\ 1 & 1 & 0 & 0 & 1 & 0 & 0 & 0 & 0 & 1 & 0 & 0 & 0 & 0 & 0 \\ 1 & 1 & 0 & 0 & 1 & 0 & 0 & 0 & 0 & 0 & 1 & 0 & 0 & 0 & 0 \\ 1 & 0 & 1 & 0 & 0 & 1 & 0 & 0 & 0 & 0 & 0 & 1 & 0 & 0 & 0 \\ 1 & 0 & 1 & 0 & 0 & 1 & 0 & 0 & 0 & 0 & 0 & 0 & 1 & 0 & 0 \\ 1 & 0 & 1 & 0 & 0 & 0 & 1 & 0 & 0 & 0 & 0 & 0 & 0 & 1 & 0 \\ 1 & 0 & 1 & 0 & 0 & 0 & 1 & 0 & 0 & 0 & 0 & 0 & 0 & 0 & 1 \end{bmatrix} \qquad (32)$$

Gradient descent is performed over 1000 epochs with learning rate $l_r = 0.01$. These simulations are not compute intensive and are easily performed on a standard modern desktop or laptop. All code for simulations and figures is available on GitHub at `https://github.com/AllenInstitute/Multipathway_NeurIPS2022`. Figure (4) shows an example of the matrix $\mathbf{K_a}$ for a specific pathway $a$, which is initialized as a uniform random matrix, becoming diagonal in the singular value space. Figure 5 shows an example of learning dynamics for networks with different depths. For the network of depth 2 (for which "pathway" is an arbitrary concept, as there is only one hidden layer), the singular values are more or less divided evenly across the pathways. This symmetry is already broken for networks of depths 3 and 4. Even by depth 7, the largest singular values are concentrated on one or the other pathway. Figure 6 shows two different initializations, demonstrating stochasticity. One can see that the largest singular values are typically in one pathway or the other, but not always the same pathway.

Importantly, our analytic results apply strictly to the linearized network. Since the standard situation with deep networks is to use some form of nonlinear activation we test our results empirically using either of Tanh or ReLU. The general trend remains the same when using these nonlinearities. The results are shown in Figures 7 and 8 for ReLU and Tanh, respectively. Simulation parameters are as above with the exception that the corresponding nonlinearity is used as an activation function following each layer of each pathway and we train for longer (as shown in each Figure).

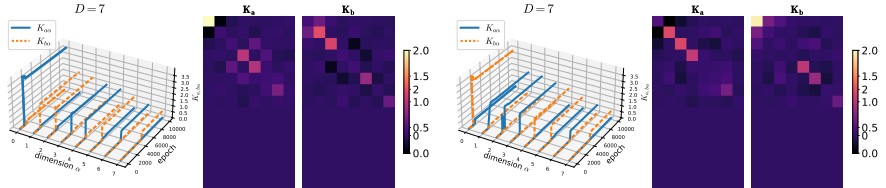

Figure 6: **Learning dynamics from different initializations in two pathway networks.** $D_a = D_b = 7$, $N_a = N_b = 1000$ and $\sigma = 0.01$. Left and right show two different seeds of the random initialization. The left side of each sub-panel shows trajectories of the matrices $\mathbf{K}_a$ and $\mathbf{K}_b$ in the singular value space. The right side of each sub-panel shows the final values for $\mathbf{K}_a$ and $\mathbf{K}_b$ in the singular value space. Note that in the different initializations, the final assignment of the features to each pathway is random.

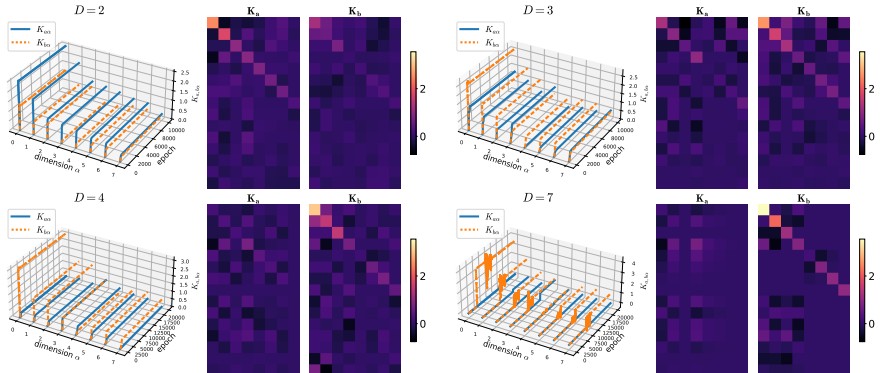

Figure 7: **Learning dynamics in two pathway networks with increasing depth, using the ReLU nonlinearity.** $D_a = D_b = D$. Left to right, top to bottom are $D = 2, 3, 4$, and $7$, respectively. The left side of each sub-panel shows trajectories of the matrices $\mathbf{K}_a$ and $\mathbf{K}_b$ in the singular value space. The right side of each sub-panel shows the final values for $\mathbf{K}_a$ and $\mathbf{K}_b$ in the singular value space. Note how with larger depths, the features have a stronger tendency to concentrate on one pathway or the other. $N_a = N_b = 1000$ and $\sigma = 0.01$.

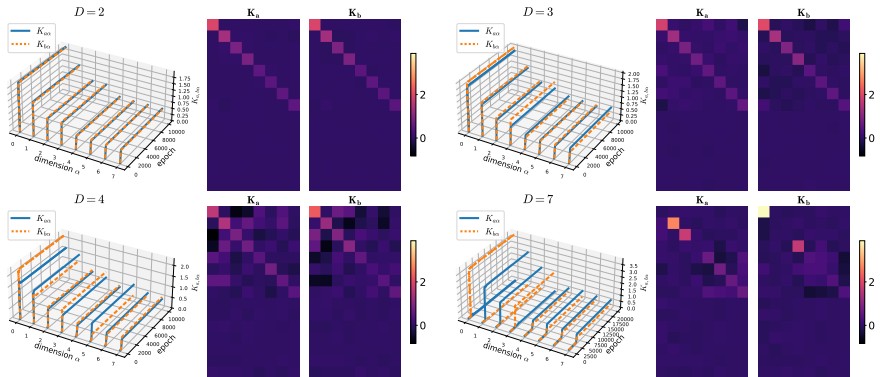

Figure 8: **Learning dynamics in two pathway networks with increasing depth, using the Tanh nonlinearity.** $D_a = D_b = D$. Left to right, top to bottom are $D = 2, 3, 4$, and $7$, respectively. The left side of each sub-panel shows trajectories of the matrices $\mathbf{K}_a$ and $\mathbf{K}_b$ in the singular value space. The right side of each sub-panel shows the final values for $\mathbf{K}_a$ and $\mathbf{K}_b$ in the singular value space. Note how with larger depths, the features have a stronger tendency to concentrate on one pathway or the other. $N_a = N_b = 1000$ and $\sigma = 0.01$.

# 5 Discussion

Many biological neural systems have several computational pathways that are thought to mediate different aspects of sensory processing, prominant examples being the "what" and "where" pathways of the macaque visual system [4, 7, 10], the "local" and "global" visual motion pathways [5], and the parallel anatomical pathways in the mouse [8]. Here we have shown, using the linearized framework of [16], that depth alone can have a profound impact upon learning dynamics in systems with multiple channels by separating features in a learned data set by pathway. It is useful to recall again the observation made for the AlexNet model [9] which featured two pathways as a result of building the architecture around two GPUs. In that case, the authors noted that each pathway learned different large scale features of the data (one largely color, the other largely texture).

We have examined the supervised case using linear regression on the output features of the network. In this framework a power iteration effect from multiple layers of computation (which can be seen in the weight matrix factors in equation 3) enhances the inherent "explaining away" effect in regression (as each pathway's features contribute to accounting for the input-output relationship) so that with increasing depth there is a tendency for each singular vector to concentrate on one pathway. Note that this does not imply that separate singular components will prefer different pathways in general. It is interesting to consider the parallels and differences between this model and the classic Oja rule that leads to PCA[13]. In that case, the learning dynamics magnifies the largest principal components of the input space and competition is introduced via weight normalization. This leads to the weights matching the largest principal components of the input data, i.e. the singular vectors of $\Sigma_x$. (This happens in a deterministic fashion, although see [12] for a higher order generalization that does not lead to deterministic selection of the largest components.) The mechanism we describe here operates in the supervised case and decomposes $\Sigma_{yx}$.

While we have used a very specific initialization scheme for mathematical purposes, we posit that (and have provided an example in which) our results are more generally applicable. In some cases this is easy to show as an extension of our work. This work is an analysis of the analytic properties of deep networks in the infinite width limit. As such, care must be taken in considering these results in the context of systems in which corrections due to finite size are large. The assumption of linearity is also important. Deep networks in practice have non-linearities. One can argue that learning dynamics will push the system to operate in a regime that maximizes the dynamic range of the response. For a function like Tanh, this should be at the inflection point. If this is the case, the network should act in a roughly linear fashion. This is just a high-level argument, however, and nonlinearities may be responsible for deviations from the results described here, although we have demonstrated in our examples that common linearities show the same empirical dynamics.

One obvious reason for having an architecture with multiple pathways is to create divergent representations for different cost functions from a common set of lower level features. In our case we consider an alternative in which all pathways subserve the same cost function. If the result is to separate features, why would such an architecture be desirable? One answer may be that multiple pathways would serve as a form of regularization. Not only does this reduce the space of model complexity, but through the mechanism we have demonstrated this can also lead to a separation of features, and perhaps more easily learnable representations for downstream tasks. This may be a useful component of pre-training procedures. It may be that multiple pathways are a source of architectural efficiency, and particularly useful when the system has few components (suggesting a reason why the mouse architecture appears more parallel than mammals with much larger brains). Quantifying these potential benefits would entail a proper model comparison across different architectures and examining performance as well as learning speed. Looking towards future work, it is important to keep in mind that biological systems may be subject to different types of cost functions, and the dynamics of those may be quite different than what we see here.

As this is an analytic approach to the theory of deep neural networks from a dynamical systems perspective, we anticipate no negative societal impacts.

## Acknowledgments and Disclosure of Funding

We thank Gabriel Koch Ocker, Blake Richards, and Shahab Bakhtiari for helpful discussions. We thank the Allen Institute founder, Paul G. Allen, for vision, encouragement, and support.

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
