# OpenReview forum: "Learning dynamics of deep linear networks with multiple pathways"
_NeurIPS.cc/2022/Conference — NeurIPS 2022 Accept_

### Official Review · Reviewer_ykV7 · 2022-07-11

**Rating:** 8
**Confidence:** 3
**Soundness:** 4 excellent
**Presentation:** 4 excellent
**Contribution:** 3 good

**Summary:**

This works aims to theoretically study the learning dynamics of deep linear networks with multiple parallel pathways (similar to parallel pathways of processing in the cortex such as dorsal/ventral, magnocellular/parvocellular pathways, etc.) in the infinite width limit. Despite the expressivity of such networks being subpar to deep learning models used for practical purposes, deep linear networks, and their associated objective functions that the authors study possess learning dynamics that show interesting nonlinear behavior. Here the authors analytically show that the different parallel pathways learn to specialize in representing different features of the training set as a function of the depth of the network and demonstrate the finding with numerical simulation. This work is exciting and is also relevant to recent self-supervised learning methods that possess parallel pathways and have been shown to mimic biological visual processing.

**Questions:**

- This is rather a comment that could be addressed by future work and not a suggested extension of the current work. The authors motivate the importance of studying the learning dynamics of deep networks with parallel pathways through their connection to biological information processing.
- However, given that there are (potentially) more biologically plausible learning rules than gradient-based error propagation (I acknowledge this area to be debated) such as local learning and Hebbian learning, one would think that studying the learning dynamics of deep networks with (or without) parallel pathways with such learning algorithms that are potentially more aligned to learning in the brain would be extremely interesting for the neuroscience community.

**Limitations:**

Yes, the authors have clearly stated the limitations of this study (restricted to deep linear networks in the infinite width limit for a particular initialization scheme, whereas commonly used deep networks in practice don’t necessarily obey all these assumptions).

**Strengths And Weaknesses:**

Strengths:
- The proposed work is an excellent extension of prior impactful work on studying the learning dynamics of deep linear networks by considering parallel pathways that are abundant in biological vision (e.g. dorsal/ventral, magno/parvocellular, etc.).
- I appreciate the well-designed figures that make it easier to grasp key concepts mentioned in the paper. I found the phase portraits in Fig. 3 to be really helpful in qualitatively interpreting the main result of the concentration of different features to different parallel pathways as a function of processing depth.
- The main results from this paper provide valuable insights into developing computational models of parallel circuits in biological neural networks.
- It is great that the authors are planning to make the code for all numerical simulations and figures available publicly on Github.
- Overall, I find the paper to be well written and organized with well-designed and clearly labeled figures wherever necessary.

Weaknesses:
- There are quite strong assumptions in place that may confine the space of models that the proposed results apply to; e.g. absence of nonlinearities, specific initialization scheme, infinite width limit.

---

> ### Author Response · Authors · 2022-08-02
> **Weaknesses**
>
> While we do indeed make strong assumptions, we suspect that our results will apply more broadly in practice.  For example, we have addressed the issue of nonlinearities with simulations on networks with Tanh and ReLU.  ReLU, in particular, severely undercuts our technical assumptions, but in practice our simulation works with that nonlinearity.  As other referees raised the same important issue, we copy our response here:
>
> This is an excellent point.  The argument for the applicability of linearized networks is that near the local optimum found by the training procedure the network will operate in a nearly linearized fashion.  For example, with the tanh nonlinearity, the system will move to the operating point where it can be most sensitive to changes in the features a given unit will respond to, which would tune units to operate near the inflection point of the nonlinearity and away from saturating values.  Of course, this is not given, and specific instances have to be checked in the absence of nonlinear analytic approaches.  As you suggest, we have tested our results using each of two common nonlinearities, Tanh and ReLU.  While the rate of convergence appears to change, the results hold in both cases, with singular values being more concentrated in a single pathway in deeper networks (we have generated analogous figures to those in 5 and 6 of the submitted manuscript; they are similar for these two nonlinearities to the linear case).  As an allied point that we mention in the introduction, AlexNet learned distinct types of features in each of its two pathways.  We feel that our mechanism is a potential explanation for this phenomenon.
>
> In our revision, we will add a discussion of the important point of applicability to nonlinearities in the main text, and also will add description of these new results, figures, and analysis, which will be added in full to supplementary material.

---

> ### Author Response · Authors · 2022-08-02
> **Response to Questions**
>
> Thank you for your review and comments, and for the intriguing suggestion for future work! As this is worth some discussion, in our revision, we will more strongly bring out these points in the main text. In particular, we will describe the suggested very interesting possibility that biological or other simpler, lower-memory learning rules will operate differently and perhaps more effectively in networks with parallel pathways, perhaps owing to the relatively few neurons and parameters that each separately contains. We are very interested in looking at the effect of multiple pathways with different cost functions and learning dynamics (see also our response to another referee regarding computational advantages for multiple pathways).

---

### Official Review · Reviewer_pqBi · 2022-07-11

**Rating:** 5
**Confidence:** 2
**Soundness:** 3 good
**Presentation:** 3 good
**Contribution:** 2 fair

**Summary:**

Inspired by mammalian brain architecture the authors the impact of having parallel pathways in artificial neural networks. They derive results analytically and conduct simulations. They show that different features of the training set typically concentrate in one of the pathways arguing for diversified representation with specialized pathways.

**Questions:**

Can you discuss more the non-linear case or perhaps conduct simulations?

How could one evaluate that diverse parallel pathways are indeed helpful in ANNs - is there some example (perhaps related to what and where streams) that could be turned into a quantifiable experiment?

**Limitations:**

I think these are overall addressed reasonably well.

**Strengths And Weaknesses:**

The paper provides both analytical results and simulation, making an interesting theoretical contribution.

To make the problem analytically tractable the authors use linear neural networks. It is not clear how the use of non-linearity would affect the results. While this might not be possible to figure out analytically, it might be possible in simulations.

The authors discuss possible utilities of diverse pathways, but the contribution would be much stronger if they would be able to show this using experiments.

It is not clear how different assumptions, such as initialization would affect the results.

---

> ### Author Response · Authors · 2022-08-02
> **Response to "How could one evaluate that diverse parallel pathways are indeed helpful in ANNs - is there some example (perhaps related to what and where streams) that could be turned into a quantifiable experiment?"**
>
> This is an excellent and important question -- thank you for raising it.
>
> It is important to consider the framework under consideration.  For example, one reading of thinking about "what" and "where" pathways is that each is governed by a specific and different cost function (with perhaps a common set of initial layers providing input). Under our framework, in which there is a common global cost function, we would like to test whether there is a computational advantage to multiple pathways vs a single monolithic pathway (with, e.g., the same number of units in each).  One natural  approach is to perform a model comparison on validation data between the two architectures, and for a range of problem definitions and data sets (some cost functions might benefit from multiple pathways, while others might not).
>
> With respect to this approach, for simple problems (like, e.g. the one we simulate in our examples, where the training error reaches zero very quickly), we do not expect a benefit from multiple pathways.  Indeed, the number of training epochs to reach zero for our example with a single pathway is *much* smaller than with two pathways; thus in this case multiple pathways not only are unnecessary to solve the problem, they slow the rate of learning as well.  We conjecture that there are tasks of sufficient complexity, depending upon the exact cost function and data set, where separating pathways does yield improved performance.  As mentioned above, one way to assess this would be via model comparison on validation data.  This particular problem does not appear to be addressable in a linearized framework.  We can analytically compute the expected validation loss on new data (drawn from the same distribution as the training data, with the same correlations).  However, the number of pathways actually drops out of the result, so we can't, e.g., assess validation accuracy vs. model complexity (judged by the number of pathways).  This is unlikely to be the case for nonlinear networks, however.  In addition, relevant to another referee's comments, different cost functions may show different, pathway-dependent results, and this is very much worth exploring.
>
>  We can consider other approaches for quantifying the benefit of multiple pathways, such as utility for pre-training for other tasks, the speed of learning, or constructing architectures for more specialized tasks on the data set.  We can construct more efficient (i.e. smaller) architectures for approximations of the same task, by removing pathways for less important singular values.  Quantifying this would entail comparing the recovered reconstruction of the output from a subset of pathways as a function of pathway size and depth.  Alternatively, we can use the identified pathways to define subtasks that make use of the singular vectors of the input/output relationship.  With a set of identified tasks, we can consider the benefit in training time gained by using the full network as part of a "pre-training" routine for these specialized tasks.  The comparison here would be the overall training time for each approach.  We conjecture there are sets of such tasks where a mulit-pathway architecture will be advantageous.
>
> We will add to our revision a discussion of these specific strategies for quantitative tests of the computational advantage of multiple pathways.  Thank you for suggesting this problem!

---

> ### Author Response · Authors · 2022-08-02
> **Response to "Can you discuss more the non-linear case or perhaps conduct simulations?"**
>
> Thank you for the review.  We respond to your first question here and the second in the next comment.
>
> Absolutely.  We appreciate this point and agree we should have included this in the initial draft.  In our revision, we will both include the requested discussion and the suggested simulations.  As this point overlaps with a point made by another referee we copy our response here:
>
> This is an excellent point.  The argument for the applicability of linearized networks is that near the local optimum found by the training procedure the network will operate in a nearly linearized fashion.  For example, with the tanh nonlinearity, the system will move to the operating point where it can be most sensitive to changes in the features a given unit will respond to, which would tune units to operate near the inflection point of the nonlinearity and away from saturating values.  Of course, this is not given, and specific instances have to be checked in the absence of nonlinear analytic approaches.  As you suggest, we have tested our results using each of two common nonlinearities, Tanh and ReLU.  While the rate of convergence appears to change, the results hold in both cases, with singular values being more concentrated in a single pathway in deeper networks (we have generated analogous figures to those in 5 and 6 of the submitted manuscript; they are similar for these two nonlinearities to the linear case).  As an allied point that we mention in the introduction, AlexNet learned distinct types of features in each of its two pathways.  We feel that our mechanism is a potential explanation for this phenomenon.
>
> In our revision, we will add a discussion of the important point of applicability to nonlinearities in the main text, and also will add description of these new results, figures, and analysis, which will be added in full to supplementary material.

---

> ### Author Response · Authors · 2022-08-08
> **Revision includes simulations with nonlinearities**
>
> We have uploaded a revised manuscript that includes figures showing the simulation results with nonlinearities.

---

### Official Review · Reviewer_8s1V · 2022-07-11

**Rating:** 6
**Confidence:** 3
**Soundness:** 3 good
**Presentation:** 3 good
**Contribution:** 3 good

**Summary:**


This paper analyzes the learning dynamics of linearized deep networks with parallel pathways driven by the same cost functions. The analysis was done from a dynamical system perspective in the tradition of Saxe et al.  on linear deep networks in the infinite width limit. An interesting conclusion is that at large depth, different pathways seem to compete and learn sharply distinct and diversified feature representations.

**Questions:**

Issues related to weaknesses should be addressed if possible.

**Limitations:**

The authors adequately addressed the limitations and potential negative societal impact.

**Strengths And Weaknesses:**

Strengths:
1. Fairly detailed and involved theoretical analysis, and the conclusion is verified by simulation results.
2. Results potentially reveal and illuminate a potential functional purpose of multiple pathways in the nervous systems.

Weaknesses:
1. The analysis is on linearized networks, with the nonlinearity removed, so whether this analysis is relevant to deep convolution neural networks and their power is open to question.
2. At some level, the linearized network seems to be behaving like a competitive network performing principal component analysis and independent component analysis, which would also arrive at similar "conclusions."

---

> ### Author Response · Authors · 2022-08-02
> **Response to Weakness 2**
>
> This is an excellent point; the connection between the dynamics we analyze and those of networks that perform PCA is worth unpacking.  Networks that perform PCA typically either require some form of asymmetric inhibition (e.g., unit 1 inhibits unit 2, while units 1 and 2 inhibit unit 3, and so on) or they will generate output in the space spanned by the PCs, rather than identify the PCs themselves (e.g. 2 output units will compute some linear combination of the first two PCs) (Oja, 1992).
>
> Thinking about PCA as arising from a basic Hebbian rule between input and output neurons, the dynamics on the weights becomes exponential and thus the largest eigenvalues will contribute the fastest dynamics.  Specifically,  $y = \vec{w}\cdot \vec{x}$ with $\Delta \vec{w} \propto y\vec{x} =( \vec{w}\cdot \vec{x}) \vec{x}$ leads to a dynamics of the form $d\vec{w}/dt = \Sigma_{xx}\vec{w}$, where $\Sigma_{xx}$ is the correlation of the input variables in $\vec{x}$.  Hence, the largest eigenvector of $\Sigma_{xx}$ dominates the dynamics and the relative contributions of others tends to zero.  This isn't "competitive" in the sense that the different eigenvectors do not suppress each other.  One sees a more explicit form of competition arising between PCs when the weight vector is normalized (say, to the L2 norm).  In this case the dynamics explicitly force the other components to zero and the steady state solution is the unique largest eigenvector (or a linear combination in the case of degeneracy).
>
> PCA is, of course, an unsupervised method whereas the problem we consider is supervised.  With the assumption that the input correlations are the identity, the solution to the regression problem requires the weights to be given by input/output correlation $\Sigma_{yx}$ (this is essentially the content of equation 12 for the steady state of the gradient descent dynamics).  This is already an important difference, because our network will be concerned with the singular values of the input/output correlations, $\Sigma_{yx}$, rather than with $\Sigma_{xx}$.  By itself this results in no decomposition, since there is nothing to separate the singular components of this correlation.  Adding the hidden layers is akin to asking the network to compute a latent representation suitable for constructing the output from the input.   The regression cost function, in the case of multiple pathways, introduces competition amongst the pathways because their net contribution, all taken together, all must contribute to "explain" the input/output relationship (e.g. the pathways must each sum to $\Omega$; this is a generic feature of regression and not anything specific to our network).  If one contributing pathway can "explain" the data, others are not needed (just as if one regressor can explain data, then other regressors will have lower weights; this is a form of "explaining away").  This feature by itself simply says that there will be an anti-correlation between different pathways in general.  When one pathway contributes more, another will contribute less.  Again, this is a general feature of regression.  The extra piece of the puzzle is the factors of ${\bf W}$ on the left and right sides of the brackets on the right hand side of equation 11.  In the diagonal solution space these turn into the powers of p and q in equation 17, which have an exponential dependence on depth.  Essentially, a kind of power iteration through repeated matrix multiplication enhances the relative contribution of each component, which combines with the natural competition from the regression.  Hence as we get to deeper and deeper architectures, the phase space curves in Figure 3 acquire larger curvature, resulting in a tendency towards a more "winner take all" solution.  Thus we arrive at something similar to PCA, in that the different singular components tend to get mapped to single pathways, but with no asymmetric inhibition terms, and, importantly, applying to $\Sigma_{yx}$ instead of $\Sigma_{xx}$, since we are in a supervised learning situation.  As it stands our network behaves somewhat similarly to the case of PCA without normalization, in that some components dominate over others but they aren't strictly sent to zero.  An interesting future direction is to ask what forms of interaction could be added to the network that have an effect similar to the normalizing term in the Oja dynamics of PCA that generate a strict "winner take all" dynamics.  Moreover, nothing in our dynamics prevents singular vectors from inhabiting the same pathway.  It is entirely possible that all singular components will migrate to one pathway and leave the remaining pathways "empty".  This isn't typically likely, but it is possible.  It is an interesting question to consider what dynamics (and their neural implementations) may cause the network to behave more like the traditional PCA.
>
> This is likely a fruitful avenue of future work and we thank the referee for raising this question.

---

> > ### Comment · Reviewer_8s1V · 2022-08-09
> > **Thanks for the responses**
> >
> > The fact that AlexNet learning two sets of features (color blobs versus gray-level Gabor) in the two GPUs is always very intriguing to me, and your result might actually provide an insight to what this is so. Thanks also for the extensive discussion on PCA.  But how about ICA -- or Foldiak's blind source separation network?   Nevertheless, a potentially very enlightening paper.

---

> ### Author Response · Authors · 2022-08-02
> **Response to Weakness 1**
>
> Thank you for the review.  We respond to weakness 1 here and weakness 2 in the next comment.
>
> This is an excellent point.  The argument for the applicability of linearized networks is that near the local optimum found by the training procedure the network will operate in a nearly linearized fashion.  For example, with the tanh nonlinearity, the system will move to the operating point where it can be most sensitive to changes in the features a given unit will respond to, which would tune units to operate near the inflection point of the nonlinearity and away from saturating values.  Of course, this is not given, and specific instances have to be checked in the absence of nonlinear analytic approaches.  As you suggest, we have tested our results using each of two common nonlinearities, Tanh and ReLU.  While the rate of convergence appears to change, the results hold in both cases, with singular values being more concentrated in a single pathway in deeper networks (we have generated analogous figures to those in 5 and 6 of the submitted manuscript; they are similar for these two nonlinearities to the linear case).  As an allied point that we mention in the introduction, AlexNet learned distinct types of features in each of its two pathways.  We feel that our mechanism is a potential explanation for this phenomenon.
>
> In our revision, we will add a discussion of the important point of applicability to nonlinearities in the main text, and also will add description of these new results, figures, and analysis, which will be added in full to supplementary material.

---

> ### Author Response · Authors · 2022-08-08
> **Revision includes simulations with nonlinearities**
>
> We have uploaded a revised manuscript that includes figures showing the simulation results with nonlinearities.

---

### Meta-Review · Area_Chair_jb3y · 2022-08-26

**Recommendation:** Accept
**Confidence:** Certain

**Metareview:**

Reviewers agree that this is an sound and well presented contribution.

**Award:**

No

---

### Decision · Program_Chairs · 2022-09-14

Accept